# Does More Expert Adjustment Associate with Less Accurate Professional Forecasts?

**Philip Hans Franses * and Max Welz**

Econometric Institute, Erasmus School of Economics, POB 1738, NL-3000 DR Rotterdam, The Netherlands;
mwelz@student.eur.nl
* Correspondence: franses@ese.eur.nl

**Abstract:** Professional forecasters can rely on an econometric model to create their forecasts. It is usually unknown to what extent they adjust an econometric model-based forecast. In this paper we show, while making just two simple assumptions, that it is possible to estimate the persistence and variance of the deviation of their forecasts from forecasts from an econometric model. A key feature of the data that facilitates our estimates is that we have forecast updates for the same forecast target. An illustration to consensus forecasters who give forecasts for GDP growth, inflation and unemployment for a range of countries and years suggests that the more a forecaster deviates from a prediction from an econometric model, the less accurate are the forecasts.

**Keywords:** professional forecasters; econometric model; expert adjustment; forecast accuracy

**JEL Classification:** C53

## 1. Introduction

Much research on professional forecasters concerns their joint predictive accuracy and their behavior relative to each other. Important studies are Batchelor (2001, 2007), Dovern and Weisser (2011), Frenkel et al. (2013), Isiklar et al. (2006), Lahiri and Sheng (2010), Loungani (2001), Capistrán and Timmermann (2009), Genre et al. (2013) and Laster et al. (1999), where the focus is on accuracy, on disagreement across forecasters, and their eventual herding behavior.

When the predictions of professional forecasters are averaged, the resulting consensus forecast is quite often reasonably accurate. At times of a crisis or turning points, however, they can be inaccurate all together. The latter may be due to their joint behavior, where herding is sometimes seen, see Laster et al. (1999). Often studied forecasters are those collected in the Survey of Professional Forecasters (SPF)[1] and those in consensus economics[2]. In this paper we will study the behavior of the consensus forecasters.

Despite an abundance of studies on professional forecasters, there is less research available to understand what it is that the professional forecasters actually do when they create their forecasts. They may or may not look at each other, and they may or may not use similar sources of information. In the present paper, we aim to address the potential consequences of whether they rely on an econometric model. In fact, our research question will concern the link between potential deviations from an individual-specific econometric model and forecast accuracy.

There is some evidence in the literature that more deviation from a model forecast, and hence a large sized adjustment, associates with lower accuracy, see Franses (2014) for a recent survey. In

---

this paper we study this conjecture for the forecasts from the consensus forecasters for various years, for the three key variables—GDP growth, inflation and unemployment, and for a range of countries. Our basic finding is that we find much evidence that more expert adjustment associates with lower forecast accuracy.

[1] https://www.philadelphiafed.org/research-and-data/real-time-center/survey-of-professional-forecasters/.
[2] https://www.consensuseconomics.com/.

Our paper proceeds as follows. In the next section we outline how we can construct measures, of the persistence and the variance of expert adjustment, from the observed forecasts. For many forecasters, we have a range of quotes for the same target variable. These forecast updates allow us to arrive at our estimates, where we need two key assumptions. The first assumption is that econometric model-based forecasts are updated only once a year, while the expert-adjusted forecast updates concern sequential months. If this holds true, then the observed updates appear to be informative for the adjustment process. Our second assumption is that the adjustment process can be described by a simple first-order autoregression. Another assumption on the model could be made too but that would make the estimation process a more complicated. Section 2 outlines our approach. Section 3 considers the forecasts, and we first present detailed results for the USA. Section 4 presents the results for other countries, where we present them is a summary. Section 5 concludes with our general finding that more deviation from the forecast from an econometric model associates with a deterioration of forecast accuracy.

## 2. Persistence and Variance of Adjustment

Consider a forecaster who gives a forecast $F$ for variable $y$ in year $T$. This forecast is given in each month $j$ in the years $T-1$ and $T$. Therefore, there are 24 forecasts for each year $T$. Note that January forecasts are special as these, for the first time, address a new calendar year. Therefore, out of the 24 forecasts monthly forecasts created across two years, we may view 22 of them as being useful updates.

To create a measure of persistence of forecast adjustment, we will rely on the updates, as we will explain next. A key assumption of our study is that a monthly forecast by a professional forecaster is the sum of an econometric model-based forecast and added intuition or expertise. We thus assume that:

$$Forecast = Model\ forecast + Adjustment.$$

Empirical evidence summarized in the survey in Franses (2014) supports that this assumption holds for a wide range of macro-economic and business forecasts. Another conclusion from that survey is that in practice it is rare that we observe both the finally adjusted forecasts and the model forecasts at the same time. In that case, one could simply evaluate the added value of intuition or expertise, by comparing the signs and size of adjustment with out-of-sample forecast performance, see Fildes et al. (2009) and Franses et al. (2011).

To elicit the sign and size of the added judgment, we make the assumption that model forecasts for annually observed variables are not updated each and every month, but that these are created only once in a year. Therefore, a plausible assumption is that:

$$F_{T|months\ in\ years\ T-1\ and\ T} = M_{T|T-1} + A_{T|months\ in\ years\ T-1\ and\ T},$$

where $F_{T|months\ in\ years\ T-1\ and\ T}$ refers to the 24 forecasts for each year $T$ created in the 24 months in years $T-1$ and $T$, where $M_{T|T-1}$ is the model forecast for year $T$ created in year $T-1$, and where $A_{T|months\ in\ years\ T-1\ and\ T}$ refers the monthly adjustment of these model-based forecasts. Hence, note that the model forecast $M_{T|T-1}$ is made only once in the year $T-1$. If we carry on with that assumption, then the forecast updates in year $T-1$ are given by:

$$F_{T|month\ j\ in\ year\ T-1} - F_{T|month\ j-1\ in\ year\ T-1} = M_{T|T-1} - M_{T|T-1} +$$
$$A_{T|month\ j\ in\ year\ T-1} - A_{T|month\ j-1\ in\ year\ T-1},$$

which becomes:

Therefore, we thus also have for year $T$ that:

$$F_{T|month\ j\ in\ year\ T} - F_{T|month\ j-1\ in\ year\ T} = A_{T|month\ j\ in\ year\ T} - A_{T|month\ j-1\ in\ year\ T}$$

Due to the special nature of January[3], we have for each year, 11 useful updates which only include the adjustments, and these run from February to December in years $T-1$ and $T$. With these, we can derive the properties of the adjustments based on the forecast updates.

To save notation, we denote a forecast update as $F_t^U = A_t - A_{t-1}$, where $t$ associates with the monthly frequency. Therefore, $A_t$ is the forecast adjustment with $t$ denoting the $t$-th forecast for a given year, in chronological order. Recall that although we have 24 forecasts for each year, only 22 of them are updates. Therefore, $t$ runs like $2, \ldots, 12, 14, \ldots, 24$. If we assume covariance stationarity across a given year's forecast adjustments, we can write $\gamma_0^A$ as the variance of $A_t$, $\gamma_1^A$ as the first-order autocovariance of $A_t$ and $\gamma_2^A$ as the second-order autocovariance of $A_t$, and we have:

$$\textit{Variance of updates}: \quad \gamma_0^U = 2\gamma_0^A - 2\gamma_1^A, \text{ and}$$
$$\textit{First} - \textit{order autocovariance of updates}: \quad \gamma_1^U = 2\gamma_1^A - \gamma_0^A - \gamma_2^A.$$

The final assumption that we now need is an assumption on the time series properties of $A_t$. We propose that a first order-autoregressive process may not be unreasonable. Therefore, suppose:

$$A_t = \rho A_{t-1} + \varepsilon_t,$$

with $0 < \rho < 1$, where the variance of the white noise process $\varepsilon_t$ is $\sigma_\varepsilon^2$. In that case, we have:

$$\gamma_0^A = \frac{\sigma_\varepsilon^2}{1-\rho^2},$$

$$\gamma_1^A = \frac{\rho\sigma_\varepsilon^2}{1-\rho^2},$$

$$\gamma_2^A = \frac{\rho^2\sigma_\varepsilon^2}{1-\rho^2}.$$

This gives;

$$\gamma_0^U = 2\gamma_0^A - 2\gamma_1^A = \frac{2(1-\rho)\sigma_\varepsilon^2}{1-\rho^2} = \frac{2\sigma_\varepsilon^2}{1+\rho},$$

and

$$\gamma_1^U = 2\gamma_1^A - \gamma_0^A - \gamma_2^A = \frac{\left(2\rho - 1 - \rho^2\right)\sigma_\varepsilon^2}{1-\rho^2} = \frac{(\rho-1)\sigma_\varepsilon^2}{1+\rho}.$$

Hence, the first order autocorrelation of the forecast updates is:

$$\frac{\gamma_1^U}{\gamma_0^U} = \frac{\rho-1}{2}.$$

When $-1 < \rho < 1$, this first order autocorrelation is negative, which is also found in for example (Clements (1997) Table 1) for GDP and inflation forecasts. From this first order autocorrelation of the

---

[3]    See Franses (2020) for a recent study on the January effect in professional forecasts.

forecast updates, we can thus obtain $\hat{\rho}$, the estimated persistence in the adjustments. Next, when we rewrite:

$$\gamma_0^U = \frac{2\sigma_\varepsilon^2}{1+\rho}$$

into

$$\sigma_\varepsilon^2 = \frac{1}{2}\gamma_0^U(1+\rho),$$

we can obtain $\hat{\sigma}_\varepsilon^2$, the estimated variance of the shocks to the adjustments.

**Table 1.** Estimation results for (1) for forecasting real GDP growth, USA, 1996–2018. Italics means significant at the 10% level.

| Forecaster | Parameter Estimates (Standard Errors) for the Variables: | | | |
| --- | --- | --- | --- | --- |
| | $\hat{\rho}$ | | $\hat{\sigma}_\epsilon^2$ | |
| JP Morgan | −0.487 | (0.270) | 6.948 | (2.060) |
| Nat Assn of Homebuilders | −0.097 | (0.368) | 7.331 | (4.162) |
| Eaton Corporation | −0.538 | (0.363) | 20.908 | (5.099) |
| Ford Motor Corp | 0.089 | (0.445) | 6.059 | (2.064) |
| The Conference Board | −0.131 | (0.332) | 9.192 | (3.709) |
| General Motors | −0.253 | (0.501) | 5.437 | (2.913) |
| DuPont | −0.686 | (0.509) | 3.829 | (3.000) |
| Fannie Mae | −0.239 | (0.362) | 5.363 | (3.430) |
| Inforum—University of Maryland | −0.479 | (0.523) | 4.887 | (2.789) |
| University of Michigan—RSQE | −0.531 | (0.422) | 4.775 | (1.534) |
| Georgia State University | −0.948 | (0.477) | 4.728 | (2.086) |

Finally, to examine how persistence in adjustment and the variance of the shocks to adjustment relate to forecast performance, we run the regression:

$$RMSPE = \mu + \alpha\hat{\rho} + \beta\hat{\sigma}_\varepsilon^2 + u \tag{1}$$

where we have a $\hat{\rho}$ and a $\hat{\sigma}_\varepsilon^2$ for each of the years, where RMSPE is the root mean squared prediction error for each of the forecasted years[4],[5], $\mu$ is an intercept and $u$ is an error term. Given the results in Franses (2014), we expect that more adjustment does not associate with better forecast performance, and hence we hypothesize that $\hat{\alpha}$ and $\hat{\beta}$ are positive and significant.

## 3. Forecasting Three Key Variables for the USA

First, we consider in detail the results for forecasting real GDP growth for the USA. We have data for the years 1996–2018 which are the years to be forecasted, which involves just 23 observations. Due to this small sample size, we will adopt a 10% significant level in our statistical analysis.

Each month there are somewhere in between 20 and 40 forecasters, and these can vary over the months. In our analysis, we will analyze only those forecasters who give forecasts in more than 80% of all months in which a forecast could have been made. For real GDP growth in the USA, we therefore analyze the forecasts of 11 professional forecasters, see the first column of Table 1.

Table 1 further reports $\hat{\alpha}$ and $\hat{\beta}$ in the regression model (1). We see from the estimation results that $\hat{\rho}$ contributes significantly in two of the 11 cases (JP Morgan and Georgia State University), but with an unexpected negative sign. Further, we see that $\hat{\sigma}_\epsilon^2$ contributes significantly and positive in nine of the

---

[4] By relying on the RMSPE measure we basically make an additional assumption and that is that the forecasters work under squared error loss. Extensions to alternative loss functions would be an interesting topic for further research.

[5] We use the realizations (source: World Bank) of the relevant variables available in May 2019.

11 cases. For real GDP growth forecasts, we thus learn that large sized shocks to adjustment associate with lower forecast accuracy.

It could now be that our results for GDP are driven by the revision process for this variable. Due to those revisions that become available throughout the year, forecasters may change their forecasts. To examine whether our findings for GDP are robust, we also consider two other key macroeconomic variables.

Table 2 presents the results for regression (1), but now for inflation. We see that $\hat{\rho}$ contributes significantly in two of the 11 cases, and now with expected positive sign. The last two columns of Table 2 indicate that the contribution of $\hat{\sigma}_\epsilon^2$ is significant and positive in 10 of the 11 cases.

**Table 2.** Estimation results for (1) for forecasting inflation, USA, 1996–2018. Italics means significant at the 10% level.

| Forecaster | Parameter Estimates (Standard Errors) for the Variables: | | | |
|---|---|---|---|---|
| | $\hat{\rho}$ | | $\hat{\sigma}_\epsilon^2$ | |
| JP Morgan | 0.128 | (0.144) | 6.243 | (2.147) |
| Nat Assn of Homebuilders | *0.498* | *(0.252)* | *3.412* | *(0.822)* |
| Eaton Corporation | −0.053 | (0.141) | *5.086* | *(2.299)* |
| Ford Motor Corp | 0.188 | (0.207) | *4.999* | *(1.504)* |
| The Conference Board | −0.028 | (0.149) | *2.008* | *(0.369)* |
| General Motors | 0.011 | (0.186) | *4.089* | *(2.016)* |
| DuPont | *0.358* | *(0.193)* | *2.348* | *(0.551)* |
| Fannie Mae | 0.069 | (0.186) | *5.973* | *(1.562)* |
| Inforum, University of Maryland | 0.032 | (0.143) | *3.980* | *(0.795)* |
| University of Michigan—RSQE | −0.139 | (0.170) | *7.339* | *(1.135)* |
| Georgia State University | 0.347 | (0.223) | 1.884 | (2.832) |

Finally, and again for the USA, Table 3 reports on the estimation results for (1) for unemployment. Here we see that $\hat{\rho}$ never contributes significantly, while $\hat{\sigma}_\epsilon^2$ does so, and positively, in seven of the 11 cases.

**Table 3.** Estimation results for (1) for forecasting unemployment, USA, 1996–2018. Italics means significant at the 10% level.

| Forecaster | Parameter Estimates (Standard Errors) for the Variables: | | | |
|---|---|---|---|---|
| | $\hat{\rho}$ | | $\hat{\sigma}_\epsilon^2$ | |
| JP Morgan | −0.188 | (0.179) | *13.289* | *(2.801)* |
| Nat Assn of Homebuilders | 0.038 | (0.287) | *25.649* | *(10.608)* |
| Eaton Corporation | −0.110 | (0.225) | *22.457* | *(6.206)* |
| Ford Motor Corp | −0.013 | (0.220) | *8.131* | *(1.791)* |
| The Conference Board | 0.077 | (0.240) | *10.225* | *(4.105)* |
| General Motors | −0.004 | (0.227) | 6.016 | (3.866) |
| DuPont | 0.053 | (0.336) | 5.730 | (5.319) |
| Fannie Mae | 0.061 | (0.169) | *9.233* | *(4.627)* |
| Inforum, University of Maryland | 0.303 | (0.239) | 9.387 | (5.669) |
| University of Michigan—RSQE | −0.298 | (0.208) | *12.719* | *(3.569)* |
| Georgia State University | 0.062 | (0.218) | 3.640 | (3.235) |

## 4. Further Results

To see to what extent the results for the USA in the previous section are representative for professional forecasters in other countries, we now turn to the analysis of the professional forecasters in the Eurozone, France, Germany, Italy, Japan, the Netherlands, Norway, Spain, Sweden, Switzerland

and the UK, again for the variables real GDP growth, inflation and unemployment. A summary of the results appear in Tables 4–6, respectively.

**Table 4.** Results for regression model (1) for other countries or regions, real GDP growth. The counts concern the number of cases with 10% significant estimation results.

| Country | Forecasters | $\hat{\rho}$ | | $\hat{\sigma}_\epsilon^2$ | |
|---|---|---|---|---|---|
| | | Positive | Negative | Positive | Negative |
| Eurozone | 14 | 0 | 1 | 14 | 0 |
| France | 5 | 0 | 1 | 5 | 0 |
| Germany | 17 | 1 | 2 | 17 | 0 |
| Italy | 6 | 0 | 1 | 6 | 0 |
| Japan | 8 | 0 | 0 | 7 | 0 |
| Netherlands | 4 | 0 | 0 | 3 | 0 |
| Norway | 2 | 0 | 0 | 2 | 0 |
| Spain | 6 | 1 | 1 | 6 | 0 |
| Sweden | 3 | 0 | 1 | 2 | 0 |
| Switzerland | 7 | 0 | 1 | 1 | 0 |
| UK | 12 | 1 | 2 | 3 | 0 |
| Total | 84 | 3 | 10 | 66 | 0 |
| | | 3.60% | 11.90% | 78.60% | 0% |

**Table 5.** Results for regression model (1) for other countries or regions, inflation. The counts concern the number of cases with 10% significant estimation results.

| Country | Forecasters | $\hat{\rho}$ | | $\hat{\sigma}_\epsilon^2$ | |
|---|---|---|---|---|---|
| | | Positive | Negative | Positive | Negative |
| Eurozone | 14 | 2 | 0 | 1 | 0 |
| France | 5 | 0 | 2 | 4 | 0 |
| Germany | 17 | 0 | 0 | 9 | 0 |
| Italy | 6 | 0 | 0 | 4 | 0 |
| Japan | 7 | 0 | 1 | 7 | 0 |
| Netherlands | 4 | 0 | 2 | 3 | 0 |
| Norway | 2 | 0 | 0 | 0 | 0 |
| Spain | 6 | 0 | 0 | 3 | 0 |
| Sweden | 2 | 0 | 0 | 2 | 0 |
| Switzerland | 7 | 0 | 1 | 6 | 0 |
| UK | 4 | 0 | 0 | 1 | 0 |
| Total | 74 | 2 | 6 | 40 | 0 |
| | | 2.70% | 8.10% | 54.10% | 0% |

**Table 6.** Results for regression model (1) for other countries or regions, unemployment. The counts concern the number of cases with 10% significant estimation results.

| Title | Forecasters | $\hat{\rho}$ | | $\hat{\sigma}_\epsilon^2$ | |
|---|---|---|---|---|---|
| | | Positive | Negative | Positive | Negative |
| Eurozone | 14 | 0 | 0 | 3 | 0 |
| France | 5 | 0 | 0 | 0 | 0 |
| Germany | 16 | 1 | 3 | 0 | 1 |
| Italy | 5 | 0 | 0 | 2 | 0 |
| Japan | 6 | 1 | 1 | 5 | 0 |
| Netherlands | 0 | | | | |
| Norway | 0 | | | | |
| Spain | 0 | | | | |
| Sweden | 0 | | | | |
| Switzerland | 0 | | | | |
| UK | 5 | 1 | 0 | 0 | 0 |
| Total | 51 | 3 | 4 | 10 | 1 |
| | | 2.70% | 7.80% | 19.60% | 2.00% |

The results in Table 4 for real GDP growth suggest that it is mainly $\hat{\sigma}_\epsilon^2$ that contributes positively to less forecast accuracy, that is in 78.6% of the 84 cases. Table 5 presents similar results for forecasting inflation, where now the fraction of cases with a positive contribution of $\hat{\sigma}_\epsilon^2$ is 54.1%. This percentage decreases even further for forecasting unemployment, as we can see from the bottom row of Table 6, where this percentage is now just 19.6%.

## 5. Conclusions

With two assumptions, one on the model forecast updates and one on the time series properties of adjustment to model-based forecasts, we could elicit the size of persistence and variance of such adjustment for professional forecasters. In our analysis of the effects of these two estimated variables on forecast accuracy, we learned that it a larger variance associates with lower forecast accuracy. Given the literature, this outcome could be expected. We find that the effects mainly concentrate on forecasting real GDP growth, which could in part be due to GDP revisions. On the other hand, we do find similar results for inflation and unemployment, although there is evidence is less strong.

An obvious limitation of our study is that we had to make two key assumptions. On the other hand, without any assumptions it seems not possible to study the behavior of the professional forecasters when they create their quotes.

**Author Contributions:** The authors contributed equally to the paper. All authors have read and agreed to the published version of the manuscript.

**Funding:** This research received no external funding.

**Acknowledgments:** We are grateful to two anonymous reviewers for their helpful comments.

**Conflicts of Interest:** The authors declare no conflict of interest.

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
