# Peer review of "Does More Expert Adjustment Associate with Less Accurate Professional Forecasts?"

_jrfm, doi:10.3390/jrfm13030044_

Round 1

Reviewer 1 Report

Comments to authors

Summary

The paper seeks to study the effects of judgemental adjustments that are plausibly made to forecasts produced by econometric models by professional forecasters. Using the dataset from Consensus Forecasts, the authors show that, assuming the annual data frequency for econometric modeling as well as stationary and persistent AR(1)-type monthly judgemental adjustment, it is possible to quantify persistence and variance of such adjustments. The key feature of the data that makes this possible is that forecasters update their forecasts monthly, while the frequency of published data is lower, so the updating is assumed to come from some sort of judgemental adjustment. The authors then investigate whether the persistence and variance of the deviation from the model forecast are correlated with the Root Mean Squared Prediction Error (RMSPE) using as examples forecasts of real GDP growth, inflation and unemployment for the US and several other countries. It is found that on average, judgemental adjustment with larger variance tend to be associated with higher RMSPE, thus presumably decreasing forecast accuracy.

Main comments

1. Given how crucial the assumptions of the paper are to the methodology and conclusions, I think they deserve more clarity, discussion and careful treatment. For example, the first assumption states that the econometric model forecast is made once a year, with monthly updates due to purely "expert adjustment". The first part of the assumption (annual data frequency versus, e.g., quarterly prevalent in empirical macroeconomic research) is somewhat easier to swallow, and could be motivated, among other things, by data publication lag (it would be nice if the authors provide some motivation here). However, I find the second part of the assumption, i.e., that all updates of the point forecast within the year are only due to judgemental adjustments, more unrealistic and potentially driving the results somewhat. The reason is that macroeconomic data, particularly GDP, are subject to realtively frequent and often sizeable (e.g., during the Great Recession) revisions. The data are typically revised quarterly, if not monthly for some countries, and not just for only the most recent datapoints - sometimes revisions concern relatively distant past. Given this, it is easy to imagine a forecaster who re-estimates his model with the updated data and posts a new point forecast during the year - completely adjustment free. However, under the assumption in the paper, this would be treated as "expert adjustment". Assuming quarterly revision, there could potentially be three data revisions and hence three purely model based forecast updates within each year. This is a sizeable fraction of total updates, and is expected to matter particularly for GDP. In fact, what the authors pick up as the significant correlation between adjustment volatility and mean squared error, could well be the effect of GDP revisions. This sounds particularly plausible since the correlation is notably weaker for unemployment and inflation which are typically subject to less drastic data revisions. I therefore feel that the authors should address this point seriously, either by providing evidence that data revisions do not play a significant role, or somehow filtering out its effect.

2. The second assumption (that the judgemental adjustments follow an AR(1)) seems incomplete - the authors obviously implicitly assume covariance stationarity, otherwise the formulas o lines 131-132 and some subsequent ones are not valid, but this is not stated in the text. This should be rectified. Additionally, it may be helpful to see a representative time series plot of such an adjustment process to let the readers engage in some "ocular econometrics" regarding the validity of the stationarity assumption and appropriateness of AR(1) as an approximation.

3. It is implicitly assumed (but not stated) that the loss function of all the forecasters is the squared loss. This may not necessarily be the case, and "less accurate" forecasts in terms of RMSE may still be optimal in some other sense. Though proper treatment of this issue is beyond the scope of this paper, it would be nice to at least have some discussion on this and perhaps a robustness check featuring risk measures of absolute loss and percentage loss, i.e., Mean Absolute Error and Mean Absolute and/or Squared Percentage Error.

4. The paper contains multiple technically inaccurate and sloppy statements. For example, on p. 6 line 237 parameters are called "numbers", and on p. 11 lines 403-404 it is stated that the null of the ADF test is no unit root, while the opposite is true. The paper also contains multiple typos and grammar errors.

Minor comments

1. It is quite unclear what data series for macro aggregates exactly are used in the paper, and how the annual measures are constructed. Also, how are the forecast errors constructed - presumably using the most recent vintage as true values? These details should be clarified.

2. The paper could benefit from a substantial proofreading, as the amount of typos, grammar errors and sloppy language hinders its readability. E.g., p. 2 line 53 and elsewhere "lesser accuracy" should be "lower accuracy"; p. 3 line 81 "A key assumption of our study is that we assume" - "we assume" is clearly redundant here; p. 4 line 108 and onwards - math displays lack punctuation etc. This is just a list of indicative examples - there are many more issues in the text.

Author Response

dear reviewer, thank you for your excellent comments.

1: We now acknowledge that part of the results for GDP could be driven by the fact that the forecasters learn about revisions. We therefore see the results for inflation and unemployment as confirmatory.

2: We do mention the assumption about covariance stationarity.

3: We now make reference to the assumption about squared error loss. We agree, thanks. 

4: There seems to be a misunderstanding here because we do not run ADF tests

Minor comments: 

yes, we use the 2019 May data as the recent vintage

and, we have checked the entire manuscript for typos and grammatical errors.  

Reviewer 2 Report

In general, I like the idea of the paper. It concerns an interesting and yet relatively unexplored topic. The authors use appropriate methods and the conclusions are supported by the data. Though, the paper should be proofread, some minor errors occur in the text. So, please polish the language. The references have to be corrected - different styles are used! 

Author Response

dear reviewer, thank you for your helpful comments.

The style for the references is now in one format

Also, we carefully checked for typos and grammatical errors. 

Round 2

Reviewer 1 Report

As this is a revision, I forego a customary summary. The paper now acknowledges the potential limitations of the research design and provides a better description of data and methodology.

Some need for proofreading work remains, particularly in punctuation of the mathematical displays.  

Author Response

Our apologies, we have added comma's and full stops where relevant, many thanks!